# Uremic Toxin Indoxyl Sulfate Promotes Macrophage-Associated Low-Grade Inflammation and Epithelial Cell Senescence

**DOI:** 10.3390/ijms24098031

**Published:** 2023-04-28

**Authors:** Andrea Ribeiro, Feiyue Liu, Matthias Srebrzynski, Simone Rother, Karina Adamowicz, Marta Wadowska, Stefanie Steiger, Hans-Joachim Anders, Christoph Schmaderer, Joanna Koziel, Maciej Lech

**Affiliations:** 1Medizinische Klinik und Poliklinik IV, Klinikum der Universität München, Nephrologisches Zentrum, Ludwig-Maximilians-Universität München, 80336 Munich, Germany; andrea.ribeiro@med.uni-muenchen.de (A.R.); feiyue.liu@med.uni-muenchen.de (F.L.);; 2Department of Nephrology, School of Medicine, Klinikum Rechts der Isar, Technical University of Munich, 81675 Munich, Germany; 3Department of Microbiology, Faculty of Biochemistry, Biophysics and Biotechnology, Jagiellonian University, 31-007 Krakow, Poland

**Keywords:** indoxyl sulfate, macrophage, CKD, uremic toxins, inflammation, senescence, inflammaging

## Abstract

In this study, we investigated the impact of the uremic toxin indoxyl sulfate on macrophages and tubular epithelial cells and its role in modulating the response to lipopolysaccharide (LPS). Indoxyl sulfate accumulates in the blood of patients with chronic kidney disease (CKD) and is a predictor of overall and cardiovascular morbidity/mortality. To simulate the uremic condition, primary macrophages and tubular epithelial cells were incubated with indoxyl sulfate at low concentrations as well as concentrations found in uremic patients, both alone and upon LPS challenge. The results showed that indoxyl sulfate alone induced the release of reactive oxygen species and low-grade inflammation in macrophages. Moreover, combined with LPS (proinflammatory conditions), indoxyl sulfate significantly increased TNF-α, CCL2, and IL-10 release but did not significantly affect the polarization of macrophages. Pre-treatment with indoxyl sulfate following LPS challenge induced the expression of aryl hydrocarbon receptor (*Ahr*) and NADPH oxidase 4 (*Nox4*) which generate reactive oxygen species (ROS). Further, experiments with tubular epithelial cells revealed that indoxyl sulfate might induce senescence in parenchymal cells and therefore participate in the progression of inflammaging. In conclusion, this study provides evidence that indoxyl sulfate provokes low-grade inflammation, modulates macrophage function, and enhances the inflammatory response associated with LPS. Finally, indoxyl sulfate signaling contributes to the senescence of tubular epithelial cells during injury.

## 1. Introduction

Chronic kidney disease (CKD) is a growing global health concern. According to recent studies, approximately 10–15% of the adult population in western countries suffers from CKD [1,2]. The prevalence of CKD is higher in older individuals, those with a history of cardiovascular disease, and individuals with diabetes [3]. CKD characterized by declined kidney function leads to an accumulation of waste products, including uremic toxins, in the bloodstream and tissues, which can contribute to the progression of the disease and further deterioration to kidney failure [4]. The burden of CKD is expected to continue to increase in the coming years, highlighting the need for effective prevention and management strategies to reduce its impact on public health [5,6].

Classification of uremic toxins in patients with CKD is based on their behavior during dialysis and physicochemical properties. According to the European Uremic Toxin Work Group (EUTox) database, there are over 100 uremic solutes/metabolites listed and the number is expected to grow [7,8,9,10]. The accumulation of uremic toxins in patients with CKD contributes to the progression of the disease and the development of various chronic health complications, such as cardiovascular disease, neurocognitive dysfunction, but also infections [11]. Exposure to high levels of uremic toxins could impair the ability of phagocytes to engulf and destroy invading microorganisms. On the other hand, low doses of uremic toxins could induce oxidative burst, a critical component of phagocytic killing in host defense. However, constant exposure to low concentrations of uremic toxins could induce low-grade inflammation and in consequence lead to a reduction in the cell surface expression of phagocytic receptors, a lower phagocyte number, or induce immunosenescence [12]. Experimental studies have proven that elevated serum levels of the metabolite uric acid increase as urinary clearance declines due to kidney dysfunction (known as hyperuricemia), which can eventually lead to uric acid crystallization in joints and cause gout [13,14] or in the kidney to cause chronic uric acid nephropathy [15]. Recent research has suggested that kynurenine and trimethylamine N-oxide (TMAO) can increase the risk of thrombosis [16,17]. In the 5/6 nephrectomy model, high levels of urea have been found to increase ROS production, causing oxidative stress in the systemic circulation [18]. Furthermore, high levels of urea can also lead to insulin resistance by increasing oxidative stress and protein O-GlcNAcylation, which impairs insulin secretion and glycolysis [19]. Similarly, p-cresyl sulfate (PCS) has also been associated with impaired glucose homeostasis, leading to hyperglycemia and altered insulin signaling in skeletal muscle [20,21]. These few examples clearly illustrate the profound impact of uremia on physiological processes.

Indoxyl sulfate is a product of protein metabolism and is formed from the breakdown of tryptophan. This small, water-soluble, uremic toxin with a molecular weight of 213 g/mol is primarily (at least 90%) bound to plasma proteins [22]. The protein-bound nature of indoxyl sulfate and other tryptophan derivatives also renders them impervious to removal by hemodialysis [23]. Consequently, concentrations of indoxyl sulfate in patients with CKD and end-stage kidney disease (ESKD) can be up to 90 times higher than the concentrations in healthy individuals [8,23]. Elevated levels of indoxyl sulfate have been associated with increased oxidative stress, inflammation, and progression of kidney disease [24]. Indoxyl sulfate has also been described as altering the gut microbiome by promoting the growth of pathogenic bacteria and increasing the risk of hospital-acquired infections [25]. Moreover, indoxyl sulfate has been shown to impair the immune system by inhibiting the activity of white blood cells, thereby increasing the risk of bacterial overgrowth [26]. The latter findings indicate the important role of indoxyl sulfate in combating pathogens since the compound can be produced in areas of infection, where increased numbers of bacteria are present. Since the microbiome, pathogens, and host share the same tryptophan substrate, multiple levels of interactions can be foreseen, from the competition of the tryptophan (commensal flora vs. pathogens) to modulation of the interaction between the host and the microbes. The latter interaction could involve interference with the activation, migration, and phagocytic abilities of immune cells and the metabolism of parenchymal cells.

Taking into account the potential high concentrations of indoxyl sulfate in tissues and its association with the microbiome and infections, it should be considered a key target for interventions reducing the progression of kidney disease, its cardiovascular consequences, and CKD-related infections. Therefore, the precise mechanisms by which indoxyl sulfate interferes with both immune and parenchymal cell function require further investigation. In this study, we report the impact of indoxyl sulfate on the function of macrophages and renal tubular epithelial cells. We provide evidence that indoxyl sulfate modulates the body’s defense mechanisms by maintaining low-grade inflammation and causing the senescence of tubular cells.

## 2. Results

### 2.1. Indoxyl Sulfate Induces Proinflammatory Transcripts in Primary Macrophages In Vitro and Modulates Their Immune Response and Metabolic Activity upon LPS Stimulation

Several studies have suggested the effects of indoxyl sulfate on macrophage function; however, its immunomodulatory role is still under-researched. We first investigated the induction of proinflammatory cytokines by low concentrations of indoxyl sulfate in macrophages and found unchanged mRNA expression of proinflammatory cytokines as well as metabolic activity in these cells (Appendix A). In further experiments, we used higher concentrations of indoxyl sulfate (60 µg/mL), which corresponds to patients with ESKD [27,28]. High concentrations of indoxyl sulfate of 60 µg/mL did not show strong, specific proinflammatory features, as evidenced by a significant increase in *Tnf-α* and *Il10* expression in macrophages (Figure 1). Furthermore, treatment of macrophages with LPS in the presence of indoxyl sulfate led to higher expression of proinflammatory factors *Tnf-α* and *Ccl2* and downregulation of *Irak-m* expression compared to the LPS-treated group (Figure 1). Interestingly, IRAK-M is known to induce the expression of negative regulators such as SOCS1, SHIP1, and A20 that control overshooting inflammation in myeloid cells to restrict tissue damage upon excessive immune response [29,30,31]. Similar results were observed with human PBMCs (Appendix A). Thus, indoxyl sulfate mediates transcriptomic changes and promotes proinflammatory activation in macrophages.

To investigate whether indoxyl sulfate affects the cell viability and metabolic activity of macrophages, we assessed their mitochondrial ability to metabolize 3-4,5-dimethylthiazol-2,5-diphenyltetrazolium bromide (MTT). The analysis showed that cells treated with high concentrations of indoxyl sulfate did not significantly change their metabolic activity. The metabolic activity of macrophages upon 48 h of incubation with LPS and indoxyl sulfate revealed that LPS-associated metabolic activity was reduced by the abundance of uremic toxin (Appendix A). The analysis of cell death did not reveal a statistically significant difference between the groups as indicated by lactate dehydrogenase (LDH) measurement. Primary macrophages turned out to be less sensitive to indoxyl sulfate and LPS-related stress than immortalized monocyte/macrophage cell lines such as human THP1 and murine J774 (Appendix A). Thus, indoxyl sulfate inhibits macrophage metabolic activity under inflammatory conditions without affecting cell death.

### 2.2. Indoxyl Sulfate Does Not Affect the Self-Limiting Nature of NF-kB-Associated Inflammatory Signaling

Macrophage homeostasis depends on extensive regulatory mechanisms that orchestrate and sequester inflammatory signals [32,33,34,35]. We hypothesized that macrophages cultured with indoxyl sulfate would display dysregulation of homeostatic transcripts under inflammatory conditions and therefore analyzed the expression of molecules that obstruct inflammatory responses. Our preliminary results, summarized in a heat map, showed that the balance of negative regulators of inflammation is not significantly disturbed by indoxyl sulfate (Figure 2). In the heat map, Z-scores were used as a scaling method for visualization. Z-scores were calculated and plotted for each gene to ensure that the expression patterns are not overwhelmed by the expression values of highly affected genes. Further investigation revealed that only one group of negative regulators was significantly expressed (above a threshold and non-template control) and induced by LPS but unaffected by indoxyl sulfate, including *A20, Mcpip1*, and *Socs3* (Figure 2). In summary, these data show that indoxyl sulfate does not change the transcript levels of homeostatic genes in macrophages.

### 2.3. Indoxyl Sulfate Induces Moderate Inflammation by Enhancing Inflammatory Cytokine Production in Macrophages

Further, we determined the cytokine production in the supernatants collected from the macrophage cultures. Interestingly, a significant difference in the levels of MCP1, TNF-α, and IL-10 between the LPS-stimulated groups (medium and pre-stimulation with indoxyl sulfate) was found in the supernatants (Figure 3). We found no statistically significant difference for IL-6, IL-12, and IFN-γ production. Stimulation with indoxyl sulfate alone significantly increased TNF levels, but did not have a significant effect on the levels of other cytokines that were tested (Figure 3). Thus, indoxyl sulfate modifies LPS-induced cytokine production and possibly induces the proinflammatory phenotype of macrophages.

### 2.4. Indoxyl Sulfate Induces ROS Production, Enhances LPS-Induced ROS Release, and Increases Mitochondrial Superoxide Production

To determine the functional importance of the proinflammatory properties of indoxyl sulfate on macrophages, we studied ROS production and their bactericidal properties. Increased oxidative stress observed in patients who suffer from CKD is associated with a proinflammatory state of the immune system [36]. To examine whether indoxyl sulfate affects ROS production in macrophages, we used a dichloro-dihydro-fluorescein diacetate (DCFH-DA) assay and assessed oxidative stress. Macrophages significantly enhanced ROS production as early as 20 minutes after treatment with indoxyl sulfate, LPS, and indoxyl sulfate with LPS (Figure 4A). Thus, indoxyl sulfate significantly induces ROS production in macrophages and enhances ROS production under inflammatory conditions. Under the same experimental conditions, we measured mitochondrial superoxide using MitoSOX Red to determine mitochondria as a major source of ROS. Consistent with the DCFH-DA data, mitochondrial superoxide levels were found to be higher in cells grown in a medium containing indoxyl sulfate and LPS (Figure 4B). Notably, under inflammatory conditions, no significant differences between the indoxyl sulfate treated and untreated groups were observed (Figure 4C).

Next, we tested the expression of oxidative stress and ER-stress-related transcripts. Consistent with other results, we observed enhanced expression of *Nrf2*, which is involved in redox homeostasis [37]. Moreover, the transcripts linked to ER stress, such as *Xbp1* and *Atf6*, were also upregulated in BMDMs stimulated with indoxyl sulfate.

Next, we evaluated the efficacy of macrophage-mediated bacterial uptake by quantifying the number of bacteria that remained associated with the cells after 24 h phagocytosis period. Interestingly, the number of live intracellular bacteria after 24 h was significantly higher in the indoxyl sulfate group (Figure 5A). Later on, we performed a bead-based phagocytosis assay to eliminate variables associated with bacterial growth and bactericidal properties of macrophages. The results of this assay demonstrated that indoxyl sulfate improves the phagocytic capability of LPS-stimulated macrophages, which indicates enhanced cell activation (Figure 5B). However, we did not observe any differences in the transmigration of macrophages toward a chemokine gradient (as shown in Appendix A). These findings suggest that indoxyl sulfate is responsible for inducing oxidative stress, mitochondrial redox reactions, and phagocytosis without affecting their migratory ability.

### 2.5. Indoxyl Sulfate Promotes Macrophage Polarization toward a Proinflammatory Phenotype

The original concept of classical and alternative macrophage polarization is associated with their role in inflammation and disease and is often linked to the balance of immune responses [38]. Therefore, we investigated the role of indoxyl sulfate in macrophage polarization to elucidate its role in this complex spatiotemporal subject. Surface expression of macrophage differentiation and activation markers were analyzed by flow cytometry and presented as the single-stain mean fluorescence intensity (MFI) in indoxyl sulfate- and LPS-stimulated BMDMs. F4/80 and CD11b markers were expressed at high levels on LPS-stimulated cells, confirming the sufficient differentiation of primary cells (Figure 6). As expected, cells also showed expression of the costimulatory molecule, CD80, mannose receptor, CD206, and MHCII. LPS-treated macrophages exhibited the typical M1-like phenotype characterized by strong induction of CD80, MHCII, and downregulation of CD206 compared to plain medium-cultured macrophages in terms of their percentage of positive cells and MFI. The cells treated with indoxyl sulfate displayed similar phenotypic features to the cells cultured in plain medium, suggesting that there were minor effects on macrophage phenotypic changes under the tested conditions. Similarly, under proinflammatory conditions, we did not observe any differences in the MFI between macrophages stimulated with both indoxyl sulfate and LPS, and those treated with LPS only, in terms of both the percentage of positive cells and MFI.

To investigate the effects of indoxyl sulfate on macrophage polarization in intact and LPS-stimulated macrophages, the cells were “gated” using F4/80 and CD11b markers. Flow cytometry analysis revealed the proportion of CD80+ (M1-like macrophage marker) and CD206+ (M2-like macrophage marker) macrophages. Indoxyl sulfate slightly increased the proportion of CD80+ cells in unstimulated macrophages compared to macrophages treated with LPS; however, indoxyl sulfate caused no major change in the proportion of CD206+ cells during the LPS challenge (Figure 7). These data indicated that indoxyl sulfate could promote M1-like macrophage polarization and that the effect is lost in LPS-treated cells due to predominant LPS signaling. The analysis reveals significant but marginal changes in the M1-like polarization of macrophages cultured with indoxyl sulfate under basal conditions. This finding was consistent with the levels of cytokines/chemokines associated with macrophage activation measured in supernatants shown above. The significantly higher levels of TNFα and MCP-1, as well as higher levels of the anti-inflammatory cytokine IL-10, indicated a mixed phenotype of indoxyl sulfate-treated macrophages. The differences in the production of the immunosuppressive cytokine IL-10 could be due to increased oxidative stress in indoxyl sulfate-treated groups since autocrine IL-10 was shown to regulate macrophage nitric oxide (NO) production [39].

Collectively, these data indicate that indoxyl sulfate has a marginal effect on macrophage polarization promoting a proinflammatory environment.

### 2.6. Indoxyl Sulfate and LPS Co-Stimulation Induce Ahr Expression

We next explored whether indoxyl sulfate could affect its endogenous aryl hydrocarbon receptor (*Ahr*) expression in macrophages as well as organic anion transporters *Oat1* and *Oat3* which have been described as mediating the uptake of indoxyl sulfate from the plasma into the cytoplasm of the cells [40,41]. Our data showed that only the combination of indoxyl sulfate and LPS significantly induces *Ahr* expression (Figure 8). Furthermore, expression of *Oat1* but not *Oat3* increased, indicating possible differences in the elimination of harmful endogenous compounds (Figure 8). The NOX complex located in the plasma membrane, which acts as a major generator of ROS in phagocytic cells and triggers a metabolic shift toward an oxidative phenotype, was also directly involved in indoxyl sulfate signaling [42]. The combination of indoxyl sulfate and LPS significantly increased the expression of *Nox4* during inflammatory conditions. Thus, indoxyl sulfate enhances *Ahr* expression and *Nox4* expression under inflammatory conditions, suggesting efficient pro-IS priming of macrophages, indoxyl sulfate uptake, and increased oxidative stress.

### 2.7. Indoxyl Sulfate Affects the Proliferation and Metabolic Activity of Renal Tubular Epithelial Cells

The complex and dynamic interplay between macrophages and parenchymal cells is a crucial aspect of the immune system’s response to injury and disease. Moreover, parenchymal cells are the functional cells of an organ that can modulate the behavior of macrophages by releasing signaling molecules and cytokines. Therefore, apart from macrophages, we investigated the response of parenchymal cells to indoxyl sulfate to gain knowledge about tissue homeostasis. Since the main type of parenchymal cells in the kidney are renal tubular epithelial cells, we considered them for our further investigations. First, we studied the impact of indoxyl sulfate on the proliferation of primary mouse renal proximal tubular cells (mTECs) and a human cell line HK2. We used 60 µg/mL of indoxyl sulfate and combined it with LPS to mimic inflammatory conditions. We incubated the cells for 72 h and observed that it suppressed serum-dependent cell proliferation and might affect cell death under inflammatory conditions (Figure 9 and Figure 10, and Appendix A–C). These findings suggest that indoxyl sulfate might be associated with poor regeneration upon injury.

### 2.8. Indoxyl Sulfate Promotes Cellular Senescence by Activating Oxidative Stress

Since senescence is a biological process characterized by a decline in cell proliferation and accumulation of cellular damage, we investigated its main marker, senescence-associated β-galactosidase (SA-β-gal) in tubular epithelial cells. Our findings demonstrate that indoxyl sulfate increased the percentage of SA-β-gal-positive cells, and this effect was even more prominent under inflammatory conditions (as shown in Figure 11A). Consequently, the size of senescent cells was also significantly increased, as measured with light microscopy (Figure 11B). Furthermore, since the association between oxidative stress and senescence is complex and bidirectional, it is expected that oxidative stress can induce senescence by causing cellular damage. Conversely, senescent cells can also contribute to oxidative stress by producing and releasing ROS and other proinflammatory cytokines (Figure 11C,D).

To determine if elevated levels of ROS also included mitochondrial ROS (mtROS), we employed the fluorescent dye MitoSOX, which specifically detects ROS generated as a by-product of mitochondrial respiration. We observed that indoxyl sulfate-induced ROS were also detected with the MitoSOX assay (Appendix A).

Senescent cells often exhibit altered mRNA splicing patterns, changes in RNA processing and translation rates, and reduced RNA synthesis. Therefore, in combination with other senescence markers such as SA-β-gal activity and increased cell size, reduced RNA turnover or concentration can contribute to identifying senescence-like changes. Our data showed a constantly lower RNA yield from all indoxyl sulfate-stimulated parenchymal cell types used in this study (Figure 12A).

Based on the data presented above, we hypothesized that indoxyl sulfate may be required to stimulate an inflammatory phenotype in renal tubular epithelial cells. We investigated the expression of major cytokines in both primary isolated TECs and HK2 cells. Since changes in the expression levels were minor in HK2 cells, we decided to use primary tubular cells for the gene expression experiments. We hypothesized that tubular epithelial cells cultured with indoxyl sulfate would display dysregulation of homeostatic transcripts under inflammatory conditions. However, our preliminary results showed that similarly to immune cells, the delicate balance of negative regulators of inflammation is not significantly disturbed by indoxyl sulfate (as shown in Appendix A). These findings suggest that indoxyl sulfate induces moderate inflammation in tubular epithelial cells. Furthermore, we did not observe significant differences in the expression of senescence and oxidative stress markers (as shown in Appendix A). Nevertheless, since the kidney has been shown to accumulate senescent cells with age [42], we decided to isolate primary tubular epithelial cells from mice aged 4–6 weeks and 6 months and stimulated them with indoxyl sulfate. We observed a significant difference in the expression of selected senescence markers upon indoxyl sulfate stimulation (Figure 12B–D). This included especially the classical hallmark of cellular senescence, *Cdkn1a* (*p21*) (Figure 12D).

Thus, indoxyl sulfate contributes to oxidative stress and senescence in renal tubular cells. This effect was more pronounced in aging cells, suggesting that indoxyl sulfate may play a critical role in elderly individuals and during inflammaging.

## 3. Discussion

CKD is associated with the accumulation of uremic toxins/metabolites in the bloodstream causing immune dysregulation (e.g., altered immune and non-immune cell functions) that contribute to various complications, including cardiovascular disease, neurocognitive dysfunction, as well as dysbiosis, and increased risk of infections and metabolic disorders [11,23,43,44,45]. Our study investigated the effects of indoxyl sulfate, a microorganism-derived uremic toxin, on macrophage function. Since the homeostasis of tissues depends not only on immune cells but also on parenchymal cells, we also investigated the effects of indoxyl sulfate on tubular epithelial cells. We report that indoxyl sulfate significantly contributes to a systemic inflammatory state by (a) activating a prooxidant response, (b) inducing a low-grade inflammatory response in macrophages, and (c) inducing tubular epithelial cell senescence that affects inflammaging.

A growing body of evidence indicates that indoxyl sulfate may significantly contribute to the progression of CKD. In vitro studies have demonstrated that indoxyl sulfate affects the biology of tubular cells, leading to increased levels of oxidative stress, inflammation, and fibrosis [46,47,48,49]. Animal models have also indicated that indoxyl sulfate can accelerate CKD progression through nephrotoxic effects [49,50]. Additionally, serum levels of indoxyl sulfate have been linked as surrogate markers of cardiovascular disease, such as intima-media thickness and pulse wave velocity in children [51], as well as diminished endothelial function in adults with CKD [52]. Our experimental results demonstrate that macrophage activation is a significant mechanism in the systemic action of indoxyl sulfate and indicate that indoxyl sulfate can induce low-grade inflammation or accelerate an existing inflammatory state. Although the effects may appear marginal, they are consistently significant, and the continuous abundance of indoxyl sulfate in tissues may significantly influence the cytokine milieu. Our findings suggest that indoxyl sulfate induces oxidative stress and affects the phagocytic capabilities of macrophages, which is in line with previous research showing that oxidative stress is prevalent in patients with CKD [53]. Adesso et al. reported that indoxyl sulfate stimulates macrophage function and enhances the inflammatory response associated with LPS, which contributes to the immune dysfunction observed in CKD patients. The authors observed a rapid and significant increase in ROS release from macrophages, reflecting the induction of an oxidative stress state [54]. Similar findings, indicating that indoxyl sulfate induces oxidative stress, have been reported for endothelial cells [55,56], vascular smooth muscle cells [47], and tubular epithelial cells [57]. Consistent with increased ROS, we observed the polarization of macrophages toward an M1-like phenotype and increased phagocytic activity. Our data contradict the effects of indoxyl sulfate on phagocytic activity in differentiated human macrophages (HL-60) reported previously [58,59]. Therefore, further investigations are needed to understand the impact of indoxyl sulfate on macrophages. Although the effects of indoxyl sulfate might vary under different conditions and activation statuses, these studies suggest that macrophages are one of the targets of indoxyl sulfate. Our study did not find statistically significant differences in macrophage apoptosis following treatment with indoxyl sulfate. Although we observed trends in apoptosis rates, the results did not reach statistical significance, suggesting that further investigation may be needed. There may be a discrepancy between our data and the results obtained for the effects of indoxyl sulfate on apoptosis in UT7/EPO cells [60]. The authors used comparable concentrations (250 μM) of indoxyl sulfate at 48 h of treatment and observed an increase in apoptosis compared to the control condition. However, when the treatment duration was reduced to 24 h, no differences were observed. In our studies, we used macrophages that could be more resistant to stress and an abundance of indoxyl sulfate. Further experiments must be conducted to reveal if UT7/EPO cells express different levels of the aryl hydrocarbon receptor than macrophages.

Inflammation is a crucial aspect of the immune system, and macrophages employ a variety of extra- and intracellular factors to regulate it [32,33,61,62,63]. This regulation is essential for maintaining stability and constancy in immune responses and tissue regeneration. To achieve this, the immune system relies on a range of modulatory mechanisms that trigger allostasis, which refers to the ability to achieve stability through change [33]. Our results show that the tested transcripts responsible for homeostasis and allostasis did not change significantly in their abundance of indoxyl sulfate. As expected, we observed upregulation of some crucial negative regulators of inflammation, such as *A20*, *Mcpip1*, or *Socs3*, upon LPS stimulation. Interestingly, the macrophage regulatory molecule *Irak-m* showed significantly decreased expression in cells stimulated with LPS and indoxyl sulfate. Previous work from our group showed that mice deficient in IRAK-M displayed a lower number of alternatively activated macrophages [29]. A lower level of IRAK-M could skew the macrophages toward a proinflammatory phenotype. This finding was consistent with the flow cytometry analysis where indoxyl sulfate triggered M1-like macrophage development. Thus, the concept of indoxyl sulfate affecting macrophages and promoting “balance” in tissue homeostasis could be essential for the function and physiology of various tissues. Indoxyl sulfate could be partially responsible for a chronic low-grade inflammatory state observed in a wide range of chronic conditions, such as metabolic syndrome (MetS), non-alcoholic fatty liver disease (NAFLD), type 2 diabetes mellitus (T2DM), and cardiovascular disease (CVD) [64,65,66]. Experimental studies have also linked low-grade inflammation to insulin resistance and suggested microbiome and microbiome-related metabolites as one of the factors affecting the development of the syndrome [67]. Rahtes et al. showed that murine monocytes persistently challenged with super-low-dose LPS can be polarized into a low-grade inflammatory state using the TRAM/Keap1-dependent mechanism [67]. Decades of research have provided extensive knowledge regarding macrophage function upon treatment with highly immunostimulatory substances. However, further research efforts are required to study the effects of subclinical low-concentration or low-stimulatory agonists. Such studies would explain the conditions required for the establishment of a low-grade inflammatory state and its effects on macrophage phenotype and function. In this context, our research could be used to elucidate if the local infections could be a source of considerable concentrations of indoxyl sulfate that change the milieu and participate in tissue remodeling and repair.

Our data demonstrate that stimulation with LPS and indoxyl sulfate together changes the expression of the indoxyl sulfate receptor *Ahr*. This indicates that the presence of this metabolite might induce greater effects during infection, and proinflammatory conditions might prime the cells and enhance their reactivity to indoxyl sulfate. However, research studies suggest that AhR represents a negative feedback mechanism that limits the strength and duration of inflammation triggered by indoxyl sulfate. For instance, AhR signaling decreases proinflammatory signals and induces differentiation of anti-inflammatory Treg cells via various mechanisms [3,68,69,70]. Since the human gut is home to a vast array of microorganisms, it is not surprising that gut microbiota can have a significant impact on host physiology through both direct cell-to-cell interactions and indirect modulation via the production of microbial metabolites. Another relevant aspect includes a need for novel biomarkers that could be used in larger populations and independent cohorts. Metabolic biomarkers such as indoxyl sulfate could be beneficial for patients [71]. They could provide insight into the metabolic status and changes occurring within the gut, kidney, and systemic circulation. A meta-analysis comprising data from 11 studies revealed that indoxyl sulfate and p-cresyl sulfate were found to be independently linked to an increased risk of cardiovascular events and mortality in patients with CKD [72]. Another study suggests that only indoxyl sulfate, and not p-cresyl sulfate signaling, can be linked to altered *miR-126* expression, which has been implicated in vascular endothelial functions, angiogenesis, and consequently the pathogenesis of CKD [73].

Chronic low-grade inflammation, also known as “inflammaging”, is a hallmark of aging and has been linked to many age-related diseases [74,75,76]. It is thought to result from a complex interplay of genetic and environmental factors, including oxidative stress, changes in the gut microbiome, and exposure to toxins. An important physiological aspect related to inflammaging is senescence, which is a state of permanent cell cycle arrest that occurs as a result of various cellular stressors, including oxidative stress, telomere shortening, and DNA damage [76]. Our results briefly introduce the concept of indoxyl sulfate-mediated senescence without deep insights regarding telomerase activity or DNA damage. We observed changes in parenchymal cells which indicated a senescence and senescence-associated secretory phenotype; however, further investigation is needed. This has been shown to drive chronic inflammation and contribute to the development of age-related diseases. Our results support findings from the literature. For instance, Niwa et al. demonstrated that indoxyl sulfate inhibited *Klotho* expression through the production of ROS and activation of NF-kB in proximal tubular cells. This induced the expression of *SA-β-gal*, *p53*, *p21*, *p16*, and retinoblastoma protein in the aorta of hypertensive rats and consequently triggered endothelial dysfunction [77]. A similar observation was made using HUVEC endothelial cells [78,79]. The authors concluded that indoxyl sulfate accelerates the progression of CKD and cardiovascular disease by inducing nephrovascular cell senescence.

In summary, the accumulation of indoxyl sulfate can have significant effects on homeostasis, leading to oxidative stress and altered immune responses. These effects highlight the importance of removing uremic toxins through dialysis or transplantation to maintain homeostasis and improve health outcomes in individuals with kidney disease. Furthermore, it is important to consider local infections as a potential source of tryptophan metabolites and their systemic effects on immune responses and a persistent low-grade inflammation state.

## 4. Materials and Methods

Generation of primary cells: Mouse tubular epithelial cells (TECs) were seeded (5 × 10^5^ cells/mL) in a 10% FCS 1% PS K1 medium in six-well plates and grown to 50% con-fluence. In brief, the mouse kidney capsule was peeled off and the kidneys were minced finely with the back of a syringe and digested with Collagenase D (working concentration 1.5 mg/mL) for 30 min at 37 °C. The digested kidneys were sieved through a 70 µm filter and centrifuged at 1500 rpm for 5 min at 4 °C, with a brake. The pellet was resuspended in 2 mL of PBS and layered very carefully on 10 mL of 31% Percoll and centrifuged at 3000 rpm for 10 min at 4 °C, without a brake. Cells were washed twice with PBS and TECs were grown from proximal tubular segments cultured in a K1 medium composed of Dulbecco’s Modified Eagle Medium supplied with 1M of Hepes (pH 7.55), 10% FCS, hormone mix (HBSS, 31.25 pg/mL PGE-1, 3.4 pg/mL T3, 18 ng/mL hydrocortisone), 9.6 µg/mL of ITSS, 20 ng/mL of EGF, and 1% PS. The medium was changed two to three days after isolation. BMDM: Bone marrow was isolated from the femur and tibia. An 18 G needle was pushed through the bottom of a 0.5 mL Eppendorf tube and put into a 1.5 mL Eppendorf tube. The bones were placed into the 0.5 mL Eppendorf tube and centrifuged at 10,000 rpm at 4 °C for 15 s. The pellet was resuspended in 1 mL of 0.155 M NH4Cl (RBC lysis buffer at room temperature) by slowly pipetting and 2 mL more was added. The mixture was kept at room temperature for 1 min. The reaction was stopped by diluting the lysis buffer with medium (10–20 mL) followed by centrifugation at 1500 rpm at 4 °C for 2 min. Cells were washed with medium and centrifuged again under the same conditions. The cell suspension was passed through a cell strainer (70 μm) and centrifuged again. The pellet was resuspended in a 1 mL medium and cells were counted. Cells were seeded in 12/6 well plates (1.5 × 10^6^/12 well plate or 3 × 10^6^/6 well plate) in 1 or 2 mL of Dulbecco’s Modified Eagle Medium supplied with 10% FCS (or mouse serum), 1% PS, and rmM-CSF at a concentration of 2 ng/mL, respectively. After 2/3 days, 1/2 mL of medium with rmM-CSF was added to the seeded cells. On day 5, the medium was removed and replaced by a fresh medium supplied with rmM-CSF. On day 7, cells were ready for stimulation. The animals were housed in accordance with international standards for the humane care and use of animals. As part of our commitment to reducing the number of animals used in research, we utilized tissue from animals that were humanely euthanized as part of approved research or breeding projects (tissue sharing). The collection of postmortem animal tissues was conducted in a registered animal facility, ensuring compliance with regulatory requirements.

Immortalized J774, THP1, and HK2 were grown in a 75 cm^2^ flask in Dulbecco’s Modified Eagle Medium containing 10% FCS and 1% PS. In the case of adherent cell lines, subcultures were prepared by scraping. For the 75 cm^2^ flasks, all but 10 mL of the culture medium was removed. Cells were dislodged from the flask with a cell scraper, aspirated, and dispensed into new flasks, in a ratio of 1:3 to 1:6. The medium was replaced two or three times a week. All in vitro experiments were performed a minimum of two independent times. Stimulation experiments were performed as indicated in the figures (stimulation time points included 4–72 h; 1–200 ng/mL LPS; 200–60 µg/mL indoxyl sulfate).

For this assay, the Phagocytosis Quantification Kit, provided by Cayman (Item Number 500290, Ann Arbor, MI, USA), was used.

Phagocytosis assay: Primary BMDMs were isolated from 3-month-old BL6 mice and cultured in DMEM medium supplemented with 10% penicillin-streptomycin and 1% FCS to a density of 1.5 million cells per well (12 well plate). On day 5, BMDMs were stimulated with LPS (2 ng/mL), indoxyl sulfate (60 µg/mL), and a 1:1 mixture of LPS (2 ng/mL) plus indoxyl sulfate (60 µg/mL) for 24 h. The next day, cells were incubated for 3 h at 37 °C, 5% CO_2_ with IgG-FITC beads to allow for bead intake. Beads with unspecified binding to the cell surface were quenched with 0.4% trypan blue solution after the incubation period. Cells with no beads and cells with beads incubated under 4 °C were used as negative controls. After BMDM collection and resuspension in FACS buffer, data were collected using flow cytometry and analyzed with FlowJo. Bactericidal assay: Primary BMDMs were isolated from BL6 mice 3 months of age 5 days before the experiment and cultured in DMEM medium supplemented with 1% FCS to a density of 1.5 million cells per well (12 well plate). On day 5, cells were stimulated with LPS (2 ng/mL), indoxyl sulfate (60 µg/mL), and a 1:1 mixture of LPS (2 ng/mL), plus indoxyl sulfate (60 µg/mL), for 24 h. One day before co-incubation, Mach1T1R *E. coli* were cultivated in 1 × LB medium overnight at 37 °C to an optical density of 600 (OD600 = 100,000,000 cells). On the day of the experiment, BMDMs and *E. coli* were co-incubated to an MOI of 1:10 at 37 °C, 5% CO_2_ in DMEM medium supplemented with 1% FCS for 4 h to allow bacterial phagocytosis. After the incubation period, wells were washed with pre-warmed PBS two times, and extracellular bacteria were killed with fresh DMEM medium supplemented with 1% FCS and 10% penicillin-streptomycin for 90 min. After the lysis of cells with distilled water, cell lysate was plated on LB agar plates to determine bacterial CFU. Transmigration assay: Primary BMDMs were isolated from BL6 mice 3 months of age 5 days before the experiment and cultured in DMEM medium supplemented with 10% penicillin-streptomycin and 1% FCS to a density of 1.5 million cells per well (12 well plate). Cells were then seeded at a density of 60,000 cells per well in the upper chamber of 24 well Corning Transwells with a pore size of 5.0 µm (CLS3421). On the day of the experiment, BMDMs were stimulated with LPS (2 ng/mL), indoxyl sulfate (60 µg/mL), and a 1:1 mixture of LPS (2 ng/mL), plus indoxyl sulfate (60 µg/mL), for 24 h. The next day, DMEM medium supplemented with 1% FCS, 10% penicillin-streptomycin, and 10 ng/mL CCL2 was added to the lower chamber and incubated for a further 24 h to allow cell transmigration. Cells were fixated with 70% ethanol and dyed with 0.2% crystal violet. The number of transmigrated cells was quantified by Image J open source v1.8.0.

Flow Cytometry: Intracellular cytokine staining was performed using the BD Cytofix/Cytoperm Kit (BD Biosciences, Franklin Lakes, NJ, USA). Cells were cultured in DMEM medium including 10% FCS and 1% PS media until they reached confluence. Cells were then centrifuged and the pellet was suspended in complete media. Macrophages were analyzed by flow cytometry using a BD FACSCanto II flow cytometer (BD Biosciences, Franklin Lakes, NJ, USA) and FlowJo v8.7 software. Differentiated macrophages were characterized using the surface markers indicated in the figures. All antibodies were obtained from BioLegend (San Diego, CA, USA). The fixable viability dye efluor780 (eBioscience, San Diego, CA, USA) was used in every sample to identify dead cells.

Real-time quantitative PCR: The SYBR Green dye detection system was used for quantitative real-time PCR using a Light Cycler 480 (Roche, Mannheim, Germany). Gene-specific primers (225 nM; Metabion, Martinsried, Germany) were used. Standard controls for genomic DNA contamination, RNA quality, and general PCR performance were included. We used the 2ΔCT method to calculate the expression of target genes for each sample based on the CT values of the reference and target genes. To calculate the 2ΔCT for each sample, we followed the following steps: (a) we calculated the ΔCT value for each sample by subtracting the CT value of the target gene from the CT value of the reference gene in that sample and (b) we calculated the 2ΔCT value for each sample by raising 2 to the power of the ΔCT value. This method allows for the calculation of the relative expression level of the target gene in each experimental sample. We did not use the fold induction in order to compare the expression of two genes within the same sample. The resulting 2ΔCT values were further analyzed using statistical tests to determine whether there were significant differences in gene expression between groups. RNA was isolated from the samples (ideally equal cell count, an equal amount of material) using the Norgen Biotek Total RNA Purification Kit (#37500; Thorold, ON, Canada) and MagNA Lyser Green Beads (Roche, Basel, Switzerland) according to the manufacturer’s instructions. Total RNA was measured using a NanoDrop ND-1000 Spectrophotometer (Thermo Fisher Scientific, Waltham, MA, USA). For cDNA conversion, we used an equal amount of RNA (500 ng or 1 µg) of total RNA and SuperScript II Reverse Transcriptase according to the manufacturer’s instructions and as described previously [32]. To inactivate the reverse transcriptase, samples were incubated again for 5 min at 90 °C. At the end of the process, samples were stored at −20 °C. The quality of total RNA was assessed using UV spectrophotometry (a ratio of absorbance at 260 nm to 280 nm around 2.0 was considered to indicate pure RNA); agarose gel electrophoresis was performed as random sampling, instead of examining every single sample to provide information on the integrity of the RNA. We used the expression levels of housekeeping genes (*Gapdh* and *18S*) to assure a consistent and comparable quality of the samples. The primer sequences (5′–3′) are in Appendix A.

Statistical analysis: Data are presented as means ± SD or box plots. The data presented in the graphs display the representative results from a minimum of 2 independent experiments. Each individual dot on the graph represents biological replicates. The Mann–Whitney U test was used for direct comparisons between single groups, i.e., wild-type and knockout cells/mice due to the small sample size and non-parametric distribution of data (D’Agostino–Pearson normality test). We used GraphPad Prism 5.0 software. The Kruskal–Wallis nonparametric test with Dunn’s multiple comparisons between datasets was used if more than two independent groups were compared. A *p*-value < 0.05 indicated statistical significance. Statistical significance was indicated as follows: *p*-value of <0.05 (*); *p*-value of <0.01 (**); *p*-value of <0.001 (***).

## Figures and Tables

**Figure 1 ijms-24-08031-f001:**
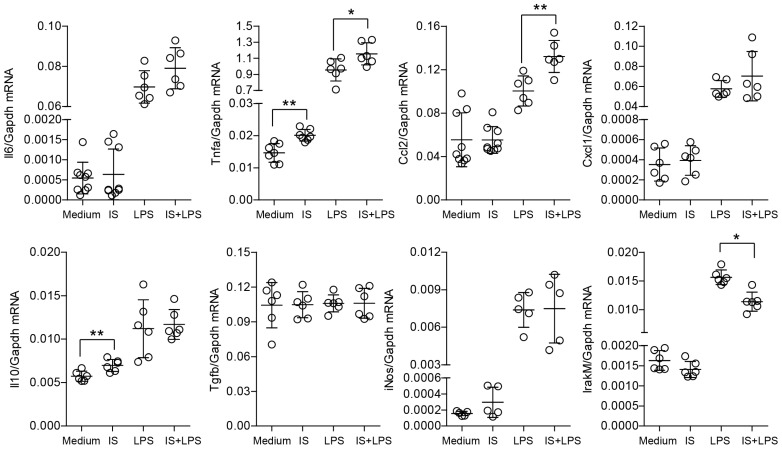
Indoxyl sulfate significantly affects the inflammatory gene expression of bone marrow-derived macrophages (BMDMs). The figure shows the mRNA expression levels of inflammation-associated genes in BMDMs, as described in the Materials and Methods Section 4. We cultured the cells in a medium containing 60 µg/mL of indoxyl sulfate (IS) and stimulated them with LPS (2 ng/mL) for 4 h. Data are shown as means ± SD; dots represent biological replicates; * *p* < 0.05, ** *p* < 0.01.

**Figure 2 ijms-24-08031-f002:**
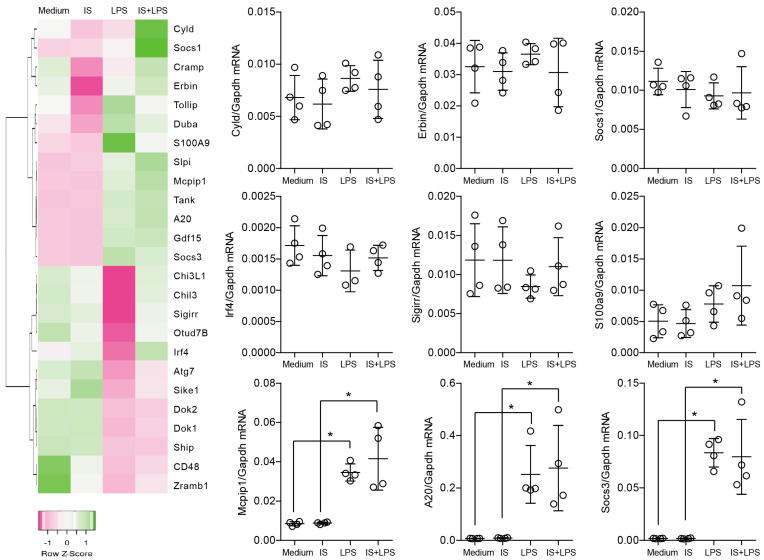
The expression levels of selected genes in BMDMs upon indoxyl sulfate (IS) and LPS treatment. The heat map shows the expression analysis of pre-selected transcripts. Genes indicated in green are upregulated and genes indicated in pink are downregulated to highlight differences between the samples. The rows are Z-score scaled for each gene separately to ensure that the expression patterns are not overwhelmed by the expression values of highly expressed transcripts. Dot plots represent the expression of selected genes and demonstrate that the distinction in color in the heat map could be a consequence of a variation in expression or low expression level. Data are shown as means ± SD; dots represent biological replicates; * *p* < 0.05.

**Figure 3 ijms-24-08031-f003:**
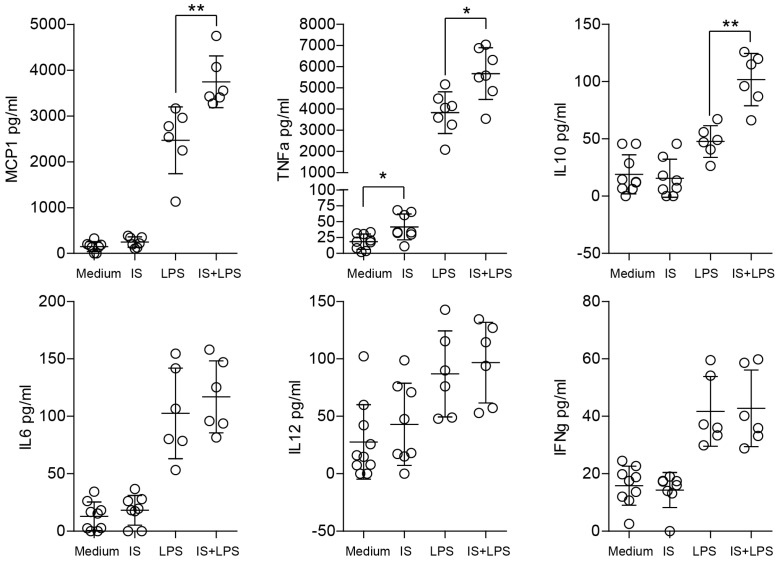
Indoxyl sulfate induces the production of inflammation-related cytokines. The secretion of inflammatory mediators was measured by a bead-based flow cytometric assay after 24 h in supernatants from bone marrow-derived macrophages (BMDMs); we cultured the cells in medium containing 60 µg/mL of indoxyl sulfate (IS) and stimulated them with LPS (2 ng/mL) for 24 h. Data are shown as means ± SD; dots represent biological replicates; * *p* < 0.05, ** *p* < 0.01.

**Figure 4 ijms-24-08031-f004:**
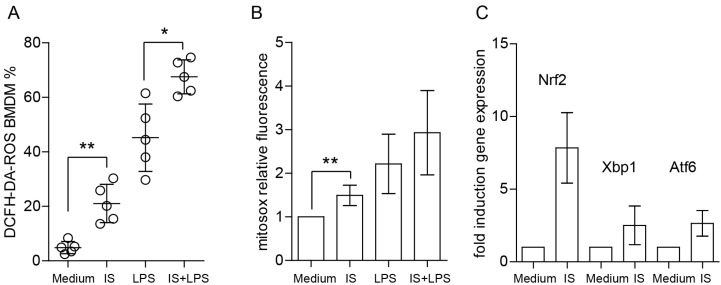
Indoxyl sulfate significantly enhances ROS production and affects redox regulation in macrophages. (**A**) BMDMs were generated, primed, and left either unstained or stained with ROS probes; dots represent biological replicates. (**B**) Mitochondrial superoxide production was detected with MitoSOX and presented as MFI fold induction (*n* = 5). (**C**) The figure shows the fold induction of mRNA expression levels of redox/ER stress-associated genes in BMDMs (*n* = 4). We cultured the cells in medium containing 60 µg/mL of indoxyl sulfate (IS) for 4 h. Data are shown as means ± SD; * *p* < 0.05, ** *p* < 0.01.

**Figure 5 ijms-24-08031-f005:**
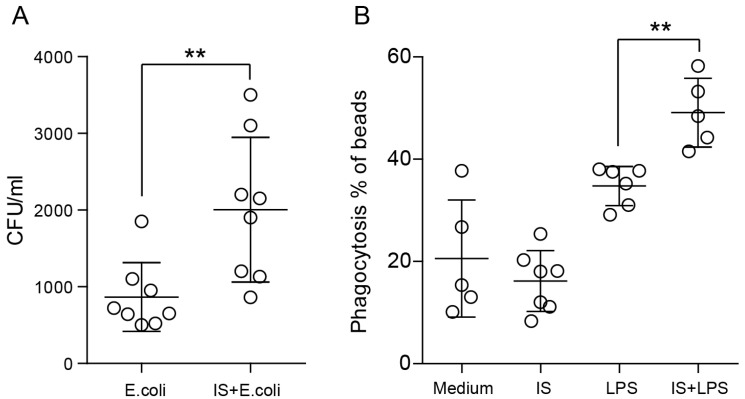
Indoxyl sulfate affects phagocytosis in macrophages. (**A**) We infected macrophages with *Escherichia coli* at an MOI of 1:10. At 24 h after infection, cell lysates were collected and plated onto LB plates for the enumeration of intracellular bacteria (CFU). (**B**) Uptake of beads by BMDMs measured by flow cytometry. Basal measurements were made in the absence of beads or at 4 °C. Three independent, representative experiments are presented. Data are shown as means ± SD; dots represent biological replicates; ** *p* < 0.01.

**Figure 6 ijms-24-08031-f006:**
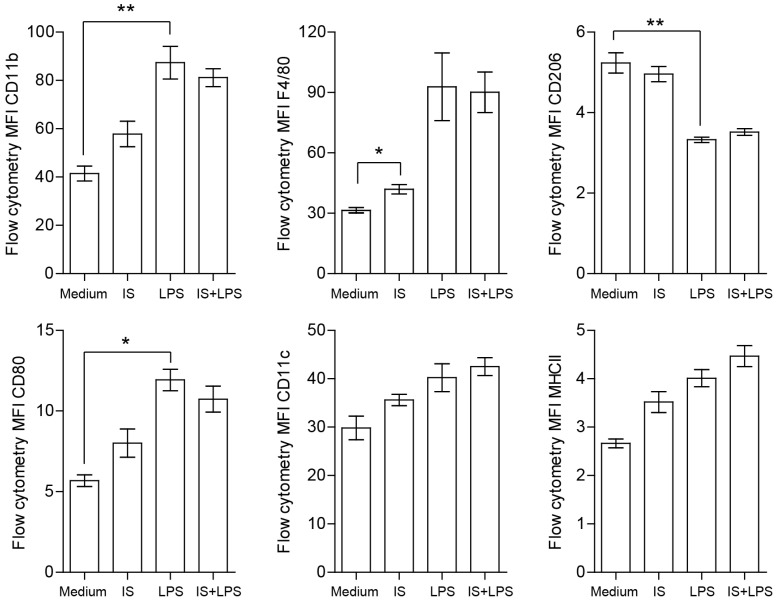
Expression of protein markers in indoxyl sulfate (IS)- and LPS-stimulated macrophages. Flow cytometry staining of surface macrophage and myeloid cell markers was detected 24 h post-stimulation under indicated conditions. Histogram plots comparing the surface expression of proteins by mean fluorescence intensity (MFI) in F480+ BMDMs treated with PBS, indoxyl sulfate, LPS, and indoxyl sulfate + LPS were quantified (*n* = 4). Data are shown as means ± SD; * *p* < 0.05, ** *p* < 0.01.

**Figure 7 ijms-24-08031-f007:**
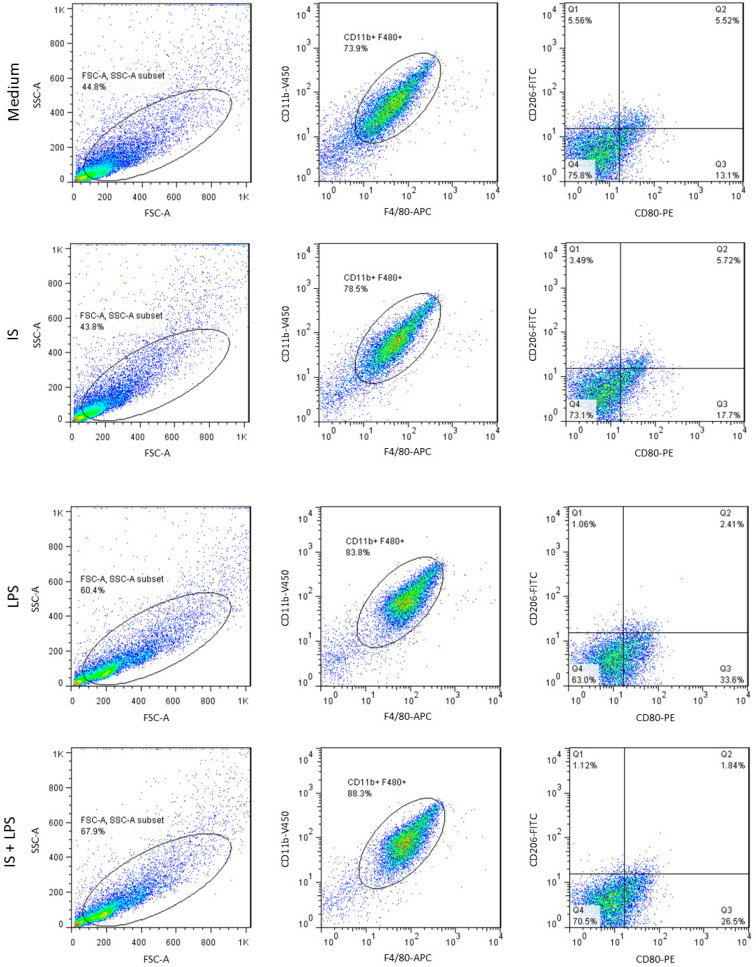
Expression of protein markers in indoxyl sulfate (IS) and LPS-stimulated macrophages. Flow cytometry staining of surface macrophage and myeloid cell markers was detected 24 h post-stimulation under indicated conditions. Flow plots correspond to the alive/CD11b/F4/80 gate. Quantification shows the proportion of M1-like and M2-like macrophages with a CD80+ (proinflammatory) or CD206+ (anti-inflammatory) phenotype at 24 h. Data are shown as means ± SD; dots represent biological replicates; * *p* < 0.05.

**Figure 8 ijms-24-08031-f008:**
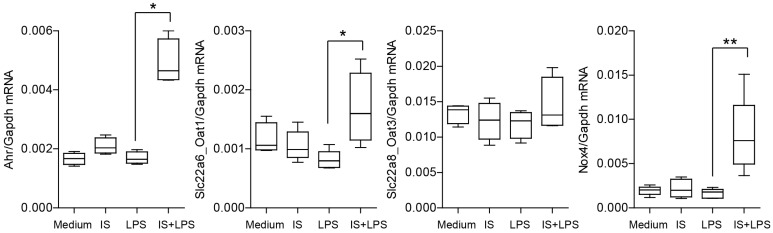
Indoxyl sulfate significantly affects the expression of *Ahr*, *Oat1*, and *Nox4* in BMDMs under inflammatory conditions. The figure shows mRNA expression levels of inflammation-associated genes in BMDMs. We cultured the cells in a medium containing 60 µg/mL of indoxyl sulfate (IS) and stimulated them with LPS (2 ng/mL) for 4 h. Data are shown as box plots (*n* > 4); * *p* < 0.05, ** *p* < 0.01.

**Figure 9 ijms-24-08031-f009:**
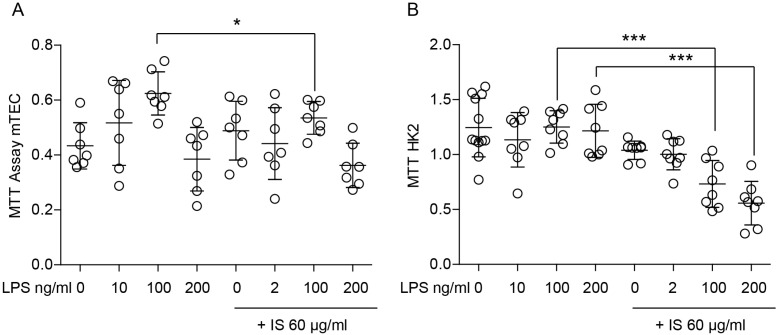
Indoxyl sulfate significantly inhibits the proliferation of tubular epithelial cells. (**A**) The viability and proliferation of mouse tubular epithelial cells (TECs) were measured using an MTT assay upon LPS stimulation (2–200 ng/mL) and in the absence or presence of indoxyl sulfate (IS). (**B**) The viability and proliferation of the human kidney epithelial cell line HK2 were measured using an MTT assay upon LPS stimulation (2–200 ng/mL) and in the absence or presence of indoxyl sulfate (IS). Data are shown as means ± SD; dots represent biological replicates; * *p* < 0.05; *** *p* < 0.001.

**Figure 10 ijms-24-08031-f010:**
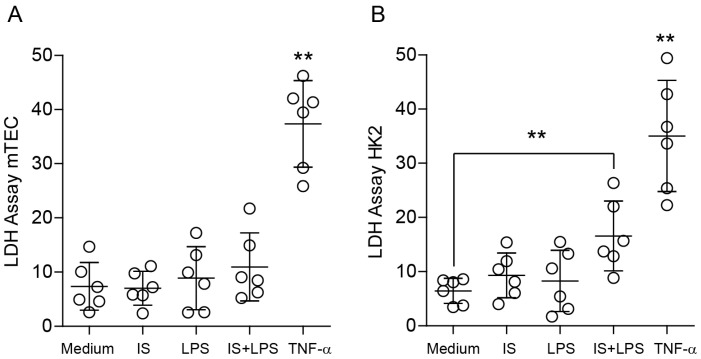
Effect of indoxyl sulfate on the viability of tubular epithelial cells. (**A**) The viability of mouse tubular epithelial cells (TECs) was measured using an LDH assay upon LPS stimulation (200 ng/mL) and in the absence or presence of indoxyl sulfate (IS). (**B**) The viability of the human kidney epithelial cell line HK2 was measured using an LDH assay upon LPS stimulation (200 ng/mL) and in the absence or presence of indoxyl sulfate (IS). Data are shown as means ± SD; dots represent biological replicates; ** *p* < 0.01.

**Figure 11 ijms-24-08031-f011:**
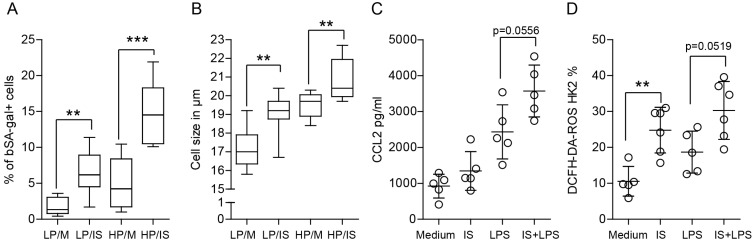
Indoxyl sulfate significantly affects cell senescence and the inflammatory response of tubular epithelial cells. (**A**) SA-β-gal expression and (**B**) cell size in low passages (LP < 10) and high passages (HP > 25) of tubular epithelial cells after the induction of cellular senescence with indoxyl sulfate. (**C**) High passage cells stimulated with indoxyl sulfate and/or LPS were investigated for signs of inflammation (CCL2 ELISA upon 24 h of stimulation and (**D**) ROS production upon 4 h of stimulation). Data are shown as box plots (*n* > 4) or means ± SD; dots represent biological replicates. ** *p* < 0.01. *** *p* < 0.001.

**Figure 12 ijms-24-08031-f012:**
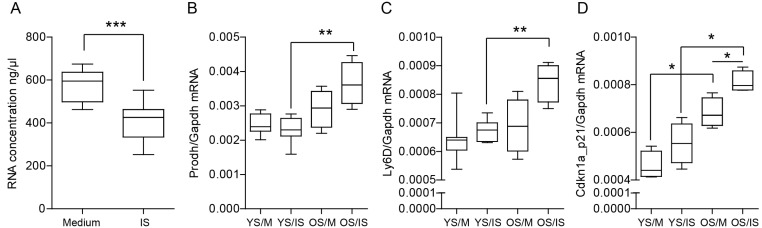
Indoxyl sulfate significantly affects the cell senescence-like phenotype of tubular epithelial cells. (**A**) The concentration of isolated RNA was constantly lower in tubular epithelial cells after stimulation with indoxyl sulfate (*n* = 9). (**B**–**D**) Box plots showing mRNA expression levels of senescence-associated genes in renal tubular cells isolated from young mice (YS) or old mice (OS). We cultured the cells in a medium containing 60 µg/mL of indoxyl sulfate (IS) for 72 h (*n* = 4). Data are shown as means ± SD; * *p* < 0.05, ** *p* < 0.01, *** *p* < 0.001.

## Data Availability

Not applicable.

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
