# Peer review of "Uremic Toxin Indoxyl Sulfate Promotes Macrophage-Associated Low-Grade Inflammation and Epithelial Cell Senescence"

_ijms, 2023, doi:10.3390/ijms24098031_

Round 1

Reviewer 1 Report

This is a very interesting manuscript by Ribero et.al, addressing role of indoxyl sulfate in uremic toxicity and inflammation and cause of kidney epithelial damage. The authors presented, invitro evidence of indoxyl sulfate induces oxidative stress and toxicity in LPS challenged epithelial cells.  Overall, manuscript was well written with report data was good with figures. However, there are a FEW PLACES in the manuscript in which the presentation must be improved as noted below.

Line 131-32: figure 1: what method was used analyze the real time data?

Label all figures with legends and indicate a, b… in figures to avoid confusion.

Line 183-184 and Line 184-185 statements are contradicting. IS didn’t  significantly induce cytokine production according to fig. 3. 

Line 256: why Indoxyl sulfate and LPS-treated macrophages displayed similar phenotypes as macrophages treated?

Line 258: figure 6: Why MFI was compared between medium and LPS?

1. Authors used more references in results section,

2. Authors explanation corresponding to some figures are missing.

Reviewer 2 Report

Authors present novel and crucial from the pathophysiological point of view study, broadly exploring effect of uremic toxin, indoxyl sulfate, on macrophages function and tubular epithelial cells senescence. Since CKD progression is tightly correlated with inflammatory status and uremic toxins related with diet/gut microbiota activity, this study adds a lot of important data helping to understand the direct effect of indoxyl sulfate on kidney cells.

Minor remarks:

1) please consider shortening the introduction and focus on indoxyl sulfate, not other uremic toxins;

2) some references do not have related numbers (line 91, line 94);

3) is 'high physiological concentration' of indoxyl sulfate equal to 'concentration corresponding to patients with ESKD'? (lines 120-123);

4) please remove is possible from Figure 4 (right) grey background on genes names;

5) please consider improving Figure's 7 resolution, first 3 graphs from flow cytometry are really hard to read;

6) please try to use abbreviations when they were already used/explained previously, e.g. SA B-gal (line 349 and 351).

Reviewer 3 Report

Ribeiro et al submit an original research article entitled "Uremic toxin indoxyl sulfate promotes macrophage-associated low-grade inflammation and epithelial cell senescence".  In this interesting work, they look at the impact of  the uremic toxin indoxyl sulfate on macrophages and tubular epithelial cells and its role in modulating the response to pro-inflammatory conditions. The authors suggest that indoxyl sulfate provokes low-grade inflammation, modulates macrophage function, and enhances the inflammatory response associated with LPS. Finally, indoxyl sulfate signaling contributes to the senescence of tubular epithelial cells during injury.

To prepare epithelial primary cells, it is indicated that the kidney capsule (of mice?) was peeled off. Also, primary BMDMs were isolated from BL6 mice 3 months before the experiment. I think it is necessary to provide an ethics agreement? Can the authors discuss this?

How was total RNA quality assessed?

the number of n (technical and biological) should be indicated in each figure legend?

The authors use really high concentrations of indoxyl sulfate, which corresponds to patients with ESK. They do not affect cell death of their blood cells. However in the same concentrations, IS triggered apoptosis in other blood cells (erythropoietic precursors see PMID 36596353) . Can the authors discuss this?

In discussion, could the authors discuss the relevance of IS levels in IRC patients in relevance to nex biomarkers (see for example PMID: 34638892, PMID: 27636773)

A final  recapitulative figure would be welcome

Round 2

Reviewer 3 Report

Changes are satisfactory